# NMR-Based Plant Metabolomics in Nutraceutical Research: An Overview

**DOI:** 10.3390/molecules25061444

**Published:** 2020-03-23

**Authors:** Giovanna Valentino, Vittoria Graziani, Brigida D’Abrosca, Severina Pacifico, Antonio Fiorentino, Monica Scognamiglio

**Affiliations:** 1Dipartimento di Scienze e Tecnologie Ambientali Biologiche e Farmaceutiche-DiSTABiF, Università degli Studi della Campania Luigi Vanvitelli, via Vivaldi 43, I-81100 Caserta, Italy; giovanna.valentino@unicampania.it (G.V.); brigida.dabrosca@unicampania.it (B.D.); severina.pacifico@unicampania.it (S.P.); 2Department of Microbiology, Tumor and Cell Biology (MTC), Biomedicum B7, Karolinska Institutet, 17165 Stockholm, Sweden; vittoria.graziani@ki.se; 3Dipartimento di Biotecnologia Marina, Stazione Zoologica Anton Dohrn, Villa Comunale, 80121 Naples, Italy

**Keywords:** metabolomics, NMR, nutraceuticals

## Abstract

Few topics are able to channel the interest of researchers, the public, and industries, like nutraceuticals. The ever-increasing demand of new compounds or new sources of known active compounds, along with the need of a better knowledge about their effectiveness, mode of action, safety, etc., led to a significant effort towards the development of analytical approaches able to answer the many questions related to this topic. Therefore, the application of cutting edges approaches to this area has been observed. Among these approaches, metabolomics is a key player. Herewith, the applications of NMR-based metabolomics to nutraceutical research are discussed: after a brief overview of the analytical workflow, the use of NMR-based metabolomics to the search for new compounds or new sources of known nutraceuticals are reviewed. Then, possible applications for quality control and nutraceutical optimization are suggested. Finally, the use of NMR-based metabolomics to study the impact of nutraceuticals on human metabolism is discussed.

## 1. Introduction

The last decades have witnessed a growing interest towards nutraceuticals and an exponential increase in the consumption of nutraceutical products [1]. Thanks to both their proven health promoting effects on one side, and to their fashionable character on the other side, they definitely won the attention of the public, pharmaceutical industries, and researchers. Therefore, due to the increasing request not only of new compounds, but also of new sources and of information on their activity, advanced techniques, and approaches are required. In this context, metabolomics offers an invaluable tool to the study of nutraceuticals at different levels.

Derived from “nutrition” and “pharmaceutical”, the term “nutraceutical” [2] is commonly confused with “functional food” because of its original definition as “food /part of food that provides health benefits”. Indeed, shortly after the nutraceutical term was coined, Zeisel defined them “as those diet supplements that deliver a concentrated form of a presumed bioactive agent from a food, presented in a nonfood matrix, and used to enhance health in dosages that exceed those that could be obtained from normal foods” [3]. Currently, nutraceuticals are bioactive ingredients extractable from food sources (e.g., plant and animal food), whose intake could be through medicinal forms (as supplement or drug-like). Moreover, foods are distinguishable in conventional foods, when they are naturally enriched in nutraceutical compounds, whereas functional foods are those fortified, enriched or enhanced in nutritional and/or nutraceutical compounds that provide health benefits beyond the provision of essential nutrients [2,4]. Phytochemicals in dietary plant sources, as well as ubiquitous compounds (e.g., phenols and polyphenols) also from non-food plants, covering a broad range of biological activities (such as antioxidant, anti-inflammatory, antimicrobic, anticancer, hypoglycaemic, cardioprotective, neuroprotective, etc.), are an endless reservoir of nutraceutical compounds [5,6,7,8]. Therefore, it is safe to assume that plant metabolome is the main target in the search for new nutraceuticals. 

Plant metabolome consists of two types of metabolites, the primary and the specialized (classically termed “secondary”). Although this classification is rather arbitrary and it is increasingly clear that there are many overlaps, primary metabolites are crosswise present in all organisms and are involved in basic essential functions for living, while specialized metabolites are species specific and essential for plant survival in environmental interactions [6]. The latter are also often endowed with a variety of biological activities that make these metabolites candidates as drugs and as health promoting factors.

The metabolome is the target of metabolomics analysis. This is an approach to the study of biological systems based on the simultaneous qualitative and quantitative analyses of the metabolites of a given biological system under a specific set of conditions [6]. As this, the approach is an extremely versatile tool, which is useful to study different phenomena and that possesses several significant advantages over traditional approaches. To date, there is no field related to natural products in which this approach has not done its successful appearance and developed as a powerful tool. Thus, no surprise that metabolomics and related technologies are also extensively applied to the study of nutraceuticals [9,10,11].

Metabolomics, especially if paired with bioactivity studies, can be useful in identifying novel nutraceuticals. Moreover, while the nutraceutical itself is sold as a pharmaceutical preparation, it is often necessary to investigate new sources of known compounds or to study the best conditions of growth, harvesting, etc. related to the production of nutraceuticals. To this end, metabolomics is increasingly being applied to profile crop varieties, improving their quality, understanding plant responses to stresses, as well as evaluating the natural variation in different tissues/organs, during growth or fruit ripening, or based on different genotypes and ecotypes, etc. [12,13]. Finally, metabolomics can also be used to study the fate and metabolism of nutraceuticals in human biological fluids [14], an often overlooked but very important aspect.

Although several analytical techniques can be used for metabolomics analysis [15], NMR (Nuclear Magnetic Resonance Spectroscopy) has some undeniable advantages especially concerning a relatively easy and rapid sample preparation and a fast and nondestructive analysis, features that make this a high-throughput, reproducible, and relatively inexpensive technique [16,17,18,19]. Furthermore, no other analytical technique reaches the same power when it comes to structural elucidation [16]. Finally, quantitative analysis is also possible by using a single internal standard [19]. The NMR experiment most widely used for these studies is ^1^H-NMR, although it is often also associated to 2D NMR for the structural elucidation of compounds [17].

The aim of the present review is to offer an overview of the potential applications of NMR-based metabolomics in the studies of nutraceuticals. Therefore, we do not mean it to be a comprehensive review of the large body of literature on the topic, but rather a selection of studies that highlight the extent of the impact of this approach to the growing field of nutraceuticals research along with suggestions of possible applications based on the success of the approach in related fields. To this end, the NMR-based metabolomics approach is shortly presented, especially in terms of the common experimental approaches. Then, examples of its use in the discovery of nutraceuticals and their sources are reported, followed by the use of metabolomics in the optimization of nutraceutical production. Finally, we describe how NMR-based metabolomics can be instrumental to the study of the metabolic fate of nutraceuticals in human body.

## 2. NMR-Based Plant Metabolomics

Metabolomics is now widely used in plant research, and far beyond. This comprehensive approach describes, in terms of metabolites, the physiological state of an organism or, more in general, of a biological system at a given moment, like a snapshot, with a high accuracy [18]. It is an extremely versatile tool that uses the chemical knowledge to interpret biological phenomena. Countless applications of this approach are virtually possible and numerous have already been published, encompassing several fields of research.

The central role of metabolomics in several fields is echoed by the amount of studies already outstandingly summarized and discussed, hence, the readers are invited to take view of the previously published reviews on the topic for more detailed information [16,17,18,19,20]. It is worth mentioning that not only metabolomics, target or untargeted, but also related approaches like metabolite profiling can be very useful. Here, only a brief overview of the methods is given, focusing on the NMR-based approach. At this point, it is important to mention that although several analytical techniques can be used in metabolomics, mass spectrometry (MS), and NMR are the most used and undoubtedly the most powerful.

The NMR technique most widely used is ^1^H-NMR. This is usually coupled with multivariate data analysis, although it is often also associated to 2D NMR (Table 1) for structural elucidation of compounds. In a ^1^H-NMR spectrum, each signal intensity is proportional to the number of protons generating it [18] and in a mixture the use of an internal standard gives virtually the chance for quantification of all of the signals (i.e., of all of the compounds detected in the extract). However, the most used approaches associate multivariate data analysis in order to extract the most significant features that are then further identified [21]. These features can be identified with reference to literature data, databases or based on extensive 2D NMR analysis of the extracts [22]. NMR is the solely analytical technique that allows the definitive de novo structural elucidation of compounds even in a mixture [16], with a few limitations concerning some stereochemistry-related issues.

It is usually possible to analyze a wide array of chemicals in a single analysis, being the solvent the only variable restricting the range of compounds. This makes NMR the ideal tool to analyze complex samples such as plants and food, known for their complexity [23,24]. Indeed, it has been shown that this technique is the most suitable in order to analyze the full range of compound classes that are recognized as nutraceuticals, with few exceptions, like the case of some classes of lipids [16,18]. Furthermore, in the last years, HR-MAS-NMR (High Resolution-Magic Angle Spinning-NMR) has also been increasingly applied [19], due to its potential of analyzing sample at the solid state, without prior extraction.

The high-throughput and reproducible characters of this technique are also noteworthy and are based on easy and fast sample preparation, as well as on the fast and reliable non-destructive data acquisition [25]. These features of NMR-based metabolomics lead to the acquisition of big data, produced in a short time and with high reproducibility.

The main drawback of using an NMR based approach, especially if compared to the other mainstream technique, MS, is linked to its relatively low sensitivity [18]. However, numerous efforts have been made in order to enhance sensitivity and resolutions. In this framework, a great advantage is given by spectrometers with ultra-high-field magnets operating at ^1^H resonance frequencies of 1.2 GHz or higher [16]. It is estimated that these instruments will allow for nanomolar detection limits [26].

The common pipeline of NMR-based metabolomics comprises several stages (Figure 1) and it has been extensively described elsewhere [27].

The design of the experiment phase is crucial in order to obtain results that are valuable both from the biological and from the statistical point of view. Once the design of experiment is set, experiments and/or sample collection can be carried out, taking care of quenching the metabolism immediately upon sampling [20,27]. It follows the sample preparation that in case of NMR-based metabolomics is usually carried out by direct extraction of the lyophilized and pulverized plant material in deuterated solvents [27]. Following centrifugation, the sample can be then directly analyzed by NMR. Then the ^1^H-NMR data are processed, bucketed, and analyzed by multivariate data analysis and other chemometric techniques, in order to determine the variables that are significant in terms of the initial biological question [20]. Finally, these variables are “translated” to NMR signals and ultimately to metabolites. In order to achieve this last point, besides the invaluable help offered by the literature and the (yet scarce) databases, 2D NMR experiments are very useful.

## 3. An Invaluable Tool in the Discovery and Characterization of Nutraceuticals and for Identification of New Sources of Nutraceuticals from the Plant Kingdom

The social trend towards a healthy diet and the consumption of nutraceuticals led to an increasing interest in the development of new products from natural sources to be used in substitution of synthetic ones. Nutraceuticals established itself as a competitive industry on global market reaching over $230 billion in 2018. BCC (Business Communications Company) Research estimated that the global nutraceutical market will reach $336 billion by 2023 [28].

The research of new compounds is steadily increasing in the last years as reported in Figure 2, with a significant driving force constituted by pharmaceutical and food industries, highly interested in the development of new nutraceuticals.

NMR-based metabolomics power to uncover activity-related compounds in complex plant matrices is a fact [29]. This approach has been proposed as an alternative to classical bioassay guided fractionation methods, giving the chance to screen a larger number of samples in a short time and starting with a lower amount of initial plant material. Paired with bioassays, it allows not only the identification of the most promising sources of bioactive compounds, but also of the metabolites responsible for the activity as shown in a number of studies covering the widest variety of biological properties [15,22,29,30]. Furthermore, metabolomics approach is useful to investigate specific classes of chemical compounds for “foodomics” and “nutriomics” [24].

It follows its applicability also to the discovery of new nutraceuticals. NMR-based metabolomics could be employed as dereplication strategy and used as a tool to speed up and simplify detection of common nutraceuticals in different plants, with information also related to their abundance. Therefore, screening by NMR-based metabolomics could be a valid alternative for identification of known compounds in new sources. In this context, it has contributed both to the identification of bioactive compounds and to their characterization [29], representing an indispensable tool unmatched for determining structures of unknown potential nutraceuticals.

Although this method is very powerful, one standing problem is the identification of nutraceuticals because of their complex chemical structures. Often, NMR spectra present overlapping signals, especially when analyzing mixtures or crude extracts. However, the vast arrays of available 2D-NMR approaches (Table 1) come at hand in this case [16]. An example is the study by Kadum et al.: they used an experimental approach based on NMR based metabolomics in order to identify the antioxidant bioactive compounds from different varieties of dates *Phoenix dactylifera* L. (Ajwa, Anbara, Piyarom, Rabbi, and Deglet Nour) [31]. The variables (i.e., chemical shift values) responsible for the activity were identified by multivariate data analysis (Principal component analysis-PCA- and Partial Least Square-Discriminant Analysis-PLS-DA) and then it was possible to identify the metabolites generating those variables based on an extensive 1D and 2D NMR analysis. The compounds responsible for the activity were identified as ascorbic, gallic and citric acids, and epicatechin.

The research in the field of nutraceuticals is not only based on the discovery of new compounds. In order to implement the production of nutraceuticals, information on the available sources is crucial. Indeed, an interesting approach is the screening of different plant samples in relation to their activity in an attempt to identify new sources or new compounds that could benefit human health.

Several studies have been published in which NMR-based metabolomics and related approaches have been used in order to find new sources of well-known and in some cases established nutraceuticals (Table 2).

Combined with multivariate data analysis, as previously mentioned, NMR is useful in complex mixture analysis in order to provide qualitative and quantitative metabolites information. Hence, the advantage that quantitative NMR-based metabolomics approach could bring is the quantitation of different compounds concurrently. A good example showing the time saving power of this approach compared to classical methods is the simultaneous quantification of L-citrulline and sugars in watermelons reported by Jayaprakasha and al. [32]. Cagliani et al. [33] adopted this approach to compare different coffee blends composition, while Turbitt et al. [34] to compare cranberry supplements to cranberry fruit powder reference standard.

## 4. Use in Quality Control and Optimization of Nutraceuticals

### 4.1. Quality Control of Nutraceuticals: Challenges and Opportunities

Quality control concerns not only the safety and authenticity, but also composition, properties and origin of the material from which nutraceuticals are obtained. It follows, that monitoring should occur at every step of production, processing, and storage, since many compounds could undergo changes at any point.

Classical approaches to this end include target analyses [81]. Unfortunately, sometimes this is not enough. The main problem with the target approach is that this only allows for determination of what is already known, i.e., the stability of the target compounds or the absence of toxic compounds that are known to be putatively present. This is where NMR-based metabolomics and especially the untargeted approach comes at hand. Indeed, it does not only detect the target compounds, but allows to have a complete overview of the chemical content of the formulation in analysis [82].

Quality control of nutraceuticals is a very important and debated point. First of all, the regulation about nutraceuticals is often not clear and most important highly variable among different countries [83]. This makes establishing clear principles and guidelines very complicated. As a consequence, the related literature is also not clear and presents significant overlapping with related fields. Hence, here we mainly discuss the potential of metabolomics for nutraceuticals quality control, taking into consideration the various important parameters and extrapolating examples from related fields.

Doubtlessly, the quality assurance of nutraceuticals, especially when in a fractionated plant mixture, is much more complex than in the case of pure drugs for many reasons, starting from the complexity of the matrix. Besides the inherent complexity of the plant matrix, it is necessary to consider that the metabolite content of plant material is highly variable, since it has been shown that plant metabolism is subjected to changes depending on the phenological stage, geographic origin, etc. [84,85]. In some cases, even diurnal fluctuations have been reported [86]. As this scenario was not already complex enough, changes can also be induced by external factors, i.e., stress, specific environmental conditions, etc. [87]. Moreover, the previously mentioned changes that can be induced at the time of harvesting and processing, as well depending on the shelf life and storage conditions, contribute to enhance the possible alteration of the plant material source of nutraceuticals. Finally, it is important to mention the use of pesticide during plant growth that could then be found in the nutraceuticals and further adulterations (toxic phytochemicals, mycotoxins, etc.,). Last, but not least, many issues are related to the extraction or preparation processes.

NMR-based metabolomics are an incomparable tool, capable of addressing each and all of the issues just mentioned. Indeed, not only is it able to analyze very complex mixtures, but thanks to its high-throughput nature, it also allows for rapid comparison of different cultivars, genotypes, and chemotypes [84,85,88]. It allows one to determine and monitor variations due to diurnal or seasonal changes in the metabolites levels [84,89] or induced by biotic and abiotic stresses or changes in growth conditions [90] or as a consequence of material sampling, handling and storage [91]. Finally, as earlier pointed out, it is the approach of choice for the detection of adulterations or contamination not only in the material of origin, but also in the nutraceutical preparation [82]. Based on these considerations, NMR-based metabolomics can play a key role in the quality control of nutraceuticals [92].

Maulidiani et al. recently used an NMR-based metabolomics approach in order to discriminate different persimmon (*Diospyros kaki* L) cultivars from Israel and South Korea [46]. They found that the discrimination was mainly due to bioactive compounds. Therefore, such a method can be used in order to evidence cultivars (as in this case) or genotypes, ecotypes, etc. that present the best profile in terms of nutraceutical content.

The method has shown its power also in determining the changes induced by processing in a study in which ground roasted coffee and instant coffee were compared by ^1^H-NMR and multivariate data analysis [93]. It was shown that instant coffee contained a remarkable increase in 5-(hydroxymethyl) furfural and carbohydrates, as well as a clear decrease in trigonelline, *N*-methylpyridinium, caffeine, caffeoylquinic acids, and 2-furylmethanol.

NMR based metabolomics is also very useful in simultaneously monitoring different level of possible variation in nutraceutical composition, as shown by a study comparing three different cultivars of onion (*Allium cepa* L.) typically cultivated in Italy [94]. The authors were able to build a model based on NMR fingerprints to discriminate the three *A. cepa* cultivars. Furthermore, they were able to follow the changes in nutraceuticals during storage.

Recently, a method using a combination on NMR and MS based metabolomics was reported by Farag et al. [95]: they compared eight commercial Senna preparations collected from several countries and proposed this method for quality evaluation of this preparation. Indeed, while sennosides are the main bioactive compounds in the preparation, they showed that there are other metabolites, in particular naphtalene glycosides, which can be used for Senna drugs’ authentication.

We must underline that the line between nutraceutical, food research, and natural product research is blurred. Indeed, many of the examples, herewith reported, are not related to the quality control of the nutraceutical itself, but rather on the source of the bioactive compound. However, this is also a crucial point in nutraceuticals research. We additionally have to say that the literature in the field is very wide and a systematic review is out of scope here.

Finally, it is important to mention that metabolomics can be also used for quality control of the final product, i.e., the pharmaceutical preparation [96].

### 4.2. Optimization of Nutraceutical Production

Due to the increasing demand for nutraceuticals, the optimization of their production is another crucial point.

Important parameters for the optimization of nutraceuticals need to be understood, among this elucidating the biosynthetic pathways is important both in the view of plant breeding strategy, or genetic manipulation or finally possible biotechnological applications. Identifying the enzymes that are responsible for the bioactive compounds and the genes that code for these enzymes offers the possibility of intervention to produce plants and consequently extracts that are richer in these metabolites. In many cases, metabolomics has been used as a tool to elucidate the biosynthetic pathway of biologically active specialized metabolites [97]. This does of course extend far beyond nutraceuticals research.

Biotechnological approaches to produce nutraceuticals might include expression/overexpression/coexpression either in the original plant or in cell cultures or in other organisms like yeasts and bacteria, more suitable for bioreactor applications. Metabolomics can be used also in the optimization of these processes [98], or coupled with genomics assisted breeding [12] aiming at increasing the production of the target molecule.

## 5. Assessing the Impact of Plant Derived Nutraceuticals on Human Metabolism

### 5.1. The Need to Provide Scientific Evidence on the Safety and Efficacy of Plant-Derived Nutraceuticals

Despite the increasing use of plant-derived nutraceuticals, the European legislation does not contemplate “nutraceuticals” as an independent food category. Nonetheless, these could be included into the regulatory framework (Directive 89/398/EEC), which was generated for the Foods for Particular Nutritional Uses (PARNUTs). PARNUTs, indeed, claim to diverge from the foodstuffs of general consumption as these are used for particular nutritional needs such as infant formulae and food for medical purposes. However, in contrast to the pharmacological agents that have to undertake a strict and well-established safety control process, the mode of action and/or the toxicity potentially exhibiting by nutraceuticals are not investigated as these are generally regarded as lacking of side effects [99].

Conversely, many concerns are recently rising with respect to the safety of nutraceuticals and botanical supplements. Ronis et al. claimed that, despite the lack of studies evaluating toxicity, accumulation of cases over time demonstrated that nutraceuticals’ uptake can result in adverse effects [100]. For instance, the metabolites of epigallocatechin gallate, which is supposed to be the active antioxidant principle of the green tea extract, might promote oxidative stress and liver injury [101]. Other doubts exist about the use of soy-derived isoflavones. Indeed, the purified form of isoflavones is known to exhibit estrogenic properties in vitro and in vivo. Accordingly, cases of endometriosis in women have been related to the consumption of soy-isoflavone supplements [102].

Hence, the increasing number of people taking nutraceuticals and herbal supplements along with the emerging cases of side effects [100] highlight the urgency to provide scientific evidence on their safety and efficacy [103].

### 5.2. Metabolomics Can Be Used as a Powerful Tool to Achieve Personalized Diagnostic and Prognostic Nutrition 

Following their intake, plant derived-nutraceuticals could be excreted from the body unmodified or, more often, are transformed by human tissues and/or gut microflora into further metabolites that can be detected in serum, urine, or feces [104].

In order to deepen our knowledge on the biological effects of nutraceuticals, it is important to take into account that, once distributed in the body, nutraceuticals and/or their derivatives could encounter numerous protein targets, interact with them with disparate specificity, and trigger different and unknown biological effects. This multi-target interaction capability pointed out that the study of a single molecule-protein interaction represents a reductionist and misleading approach, while omics-based technologies allow the analysis of the systemic effects of nutraceuticals [103].

Metabolomics provides a snapshot of the human small molecule metabolites, and more important, is able to detect the changing flux of metabolites in response to dietary changes of the host [105]. In fact, metabolomics coupled with chemometrics tools and specialized software can reveal biomarkers or metabolite patterns of human biological samples prior and after the intake of nutraceuticals [99].

The importance of metabolomics in nutrition has been officially recognized through the creation of the Food Biomarker Alliance (FOODBALL), which is composed of 22 partners from 11 countries and it is supported by the EU Joint Programming Initiative “A Healthy Diet for a Healthy Life”. FOODBALL project aims at discovering new biomarkers for food intake to establish a more reliable analytic tool respect to those that are commonly used in human dietary assessment such as food frequency questionnaires, food diaries, and 24 h recall methods. Importantly, to achieve the goal proposed in the project, FOODBALL will apply metabolomics as main omics tool. Moreover, PhytoHub, a freely electronic database available in the FOODBALL online platform (foodmetabolome.org/foodball), provide information regarding dietary phytochemicals and their human and animal metabolite-derivatives, including a library of mass spectrometry data.

### 5.3. NMR-Based-Approaches in Human Nutritional Metabolomics Studies

Despite the recently recognized importance of metabolomics in the field of human nutrition, the characterization of the nutritional metabolic phenotype remains an ambitious goal to achieve. First, studies were addressed to figure out the fate and the effects of phytochemicals, and botanical supplements must consider either the variation of nutrient bio-accessibility during digestion or the biotransformation processes through which food undergo in human body. Accordingly, in his review, Capozzi pointed out the existence of a food intrinsic metabolome, a gastro-intestinal derived metabolome as well as a food induced-endogenous metabolome [106].

Moreover, it is worthy to note that phytochemicals and their derivatives are hidden in the background of thousands of endogenous human metabolites, which are present in an extreme wide concentration range in biological fluids. For this reason, phytochemical detection and identification/quantification represent difficult tasks to perform. As it was in detail discussed by Van Duynhoven et al. in their review, the technical developments made in NMR instrumentations and software tools (standardization protocols, hyphenation, accurate quantification, spectral deconvolution, and high-throughput automation) again placed NMR-based approaches in a central position in nutritional metabolomics studies [107]. However, the low NMR sensitivity has greatly limited its applications so far, while MS based-approaches have been more commonly used to detect the low abundant metabolite present in human metabolome [108]. Herein, we briefly discussed how NMR technical implementations contributed to deepen our understanding into the fate of phytochemicals with nutraceutical perspectives in human body, and to detect, identify/quantify low abundant metabolites derived from their uptake.

### 5.4. Following the Intricate Fate of Nutraceuticals in the Human Body

As it was underlined above, the nutraceuticals’ biotransformation is a key step of the nutritional investigation, especially when this aims at following their fate in the human body and at detecting analogues or derivatives in complex biological fluid mixtures. Recently, to our knowledge, the best well-characterized study in this field looks into the bioconversion of polyphenols by the gut microbiota and human tissues [109].

Plant polyphenols are phytochemicals that are claimed to promote various beneficial effects on human health. In vitro studies demonstrated their anti-oxidant capabilities in inflammation processes [110] and protective effects against cardiovascular diseases [111]. However, these works did not consider the polyphenols bio-availability and -conversion. Some authors argued that the actual polyphenols availability in human tissues is far too low to justify the claimed biological effects, while others showed that polyphenols are mainly absorbed in the intestine as monomeric forms, which are further transformed in blood and urines. In the review of Van Duynhoven, it has been highlighted how the bidirectional interactions between the gut microbiota and the human host result in a co-metabolome, which is crucial for understating the complex metabolic fate of dietary polyphenols. The flavonoid naringenin was proposed to be converted in 3-(4-hydroxyphenyl) propionic and 3-phenylpropionic acid, whose mixture is absorbed from the colon and recovered in urine as such or in conjugated forms. In some cases, the aforementioned mixture could also go to β-oxidation and glycination in the liver, finally resulting in hippuric and 4-hydroxyhippuric acid. Importantly, Van Duynhoven et al. also described a nutrikinetik signature of polyphenolic intake and discussed how this could be important to map individual interactions between gut microbiota and host metabolism [109]. Moreover, Van Duynhoven performed also a nutrikinetic analysis following 48 h after polyphenol-rich black tea intake. Specifically, ^1^H-NMR profiles of urine samples coming from 20 healthy volunteers were used to elaborate a PLS-DA analysis, which in turn gave discriminant biomarkers of polyphenol dietary exposure. These metabolites mainly include hippuric acid, 4-hydroxyhippuric acid and 1,3-dihydroxyphenyl-2-*O*-sulfate, which have been suggested to derive from the microbial fermentation of polyphenols in the gut. In this study, it is also worthy to note that variations in urinary excretion of the volunteers have been detected and exploited in an attempt to provide a personalized metabolic phenotype of the tested population [112].

### 5.5. Analyzing Low Abundant Metabolites Derived from Phytochemicals Uptake

The intrinsic low sensitivity of NMR spectroscopy has been partially overcome by the implementation of NMR instrumentation and by the combination with chromatographic techniques. Single and multiple microcoil capillary flow probe NMR provided higher spectra quality respect to those deriving from traditional methods, and importantly, it led to the most remarkable improvement in the sensitivity of this technique [105]. However, to our knowledge, applications of these instrumentations/approaches for the analysis of the nutritional metabolic phenotype of human biological fluids are not yet available.

On the other hand, NMR hyphenation with Liquid chromatography (LC-NMR) and solid phase extraction (SPE-NMR) has already provided sub-metabolic profiles in which even low-abundance compounds, present in much lower orders of magnitude than the human endogenous metabolic background, can be unambiguously identified and accurately quantified [107]. A clear example of these approaches was provided by the work of Jacobs et al. In this human intervention study, the authors combined automated solid-phase extraction (SPE) with NMR metabolite profiling to investigate healthy volunteers’ consumption either of capsules including a polyphenol-enrich mixture of red wine and grape juice extract or the same polyphenol mixture dissolved in a soy drink over a period of five days. SPE-NMR allowed them to obtain three sub-metabolomes deriving from the aqueous, hydro-alcoholic, and methanol fractions of the urine parental mixture, which included polar, semi-polar, and aromatic metabolites, respectively. Results from this study revealed that capsule as well as soy drink consumption triggers an enhanced urinary excretion of 4-hydroxyhippuric acid, hippuric acid, 3-hydroxyphenylacetic, homovanillic and 3-(3-hydroxyphenyl)-3-hydroxypropionic acid [113].

### 5.6. Evaluating the Impact of Nutraceutical Compounds on Human Metabolism and Health

We above highlighted the key role of NMR-based metabolomics in analyzing how human metabolism and the gut microbiota bio-convert phytochemicals in other molecules. Furthermore, we pointed out that these human metabolites derived from dietary phytochemicals are important to reveal biomarkers of nutraceutical intake.

In this last paragraph, our aim is to shed light on the appealing possibility, yet so far still unexplored, to investigate the impact of phytochemicals on human metabolism and health by using NMR-based metabolomics.

The majority of human interventions studies in nutrition were addressed to understand the putative beneficial effects of phytochemicals in preventing and impairing the risk of cardiovascular diseases (CVD). In these studies, the monitoring of certain physiological biomarkers (e.g., blood pressure and flow-mediated dilation of blood vessels) along with the analysis of blood biochemical parameters (e.g., pro- and anti-inflammatory cytokines and lipid profiles) emerged as non-invasive tools to evaluate the health-promoting effects of dietary phytochemicals [114].

Dysregulations of lipid and lipoprotein metabolism are a crucial aspect in the development of CVD. Lipoproteins act as lipid carriers and are supramolecular protein-lipid aggregates, whose size and composition vary according to their specific function. Chylomicrons are the largest lipoproteins that mainly exist in the postprandial state, while very-large-density lipoproteins (VLDL) are smaller than chylomicrons and transport lipids in the fasting state. Following by the release of lipids to the tissues, VLDLs are transformed into smaller intermediate-density (IDL) and low-density lipoproteins (LDL). Furthermore, uptake of lipids from tissues is allowed by the high-density lipoprotein (HDL) particles. In healthy people, there is a strict equilibrium between release and uptake of lipids, while alterations of this balance have been identified in individuals with CVD risk. The analysis of LDL and HDL are common in clinics and an increased LDL/HDL ratio has been interpreted as an indicator of cardiovascular risk. However, it has been claimed that a more comprehensive analysis, which should include VLDL, IDL, LDL, and HDL subclasses, is a better risk indicator for CVD [115].

Erkkilä et al. carried out a pioneering human intervention study, which was addressed to understand whether a diet rich in fish oil could positively affect the lipoprotein profile of individuals suffering from CVD. In this study, the authors analyzed the lipoprotein subclasses and their lipid components by using NMR. Specifically, 33 patients with coronary heart disease were randomly subdivided in three groups, which followed a fatty fish (*n* = 11), a lean fish (*n* = 12), or a control (*n* = 10) diet regime for 8 weeks. The control group consumed lean beef, pork, and chicken. As a result, they showed that concentrations of n-3 fatty acids and docosahexaenoic acid was higher in the fatty fish group respect to the others. Moreover, the concentrations of free cholesterol, cholesterol esters, and total lipids in HDLs as well as the size of HDL particles was found increased in the fatty fish group. Finally, this study demonstrated that fatty fish intake at least four times per week augments HDL particle size, which could exert beneficial effects in patients suffering from coronary heart disease [115].

Apart from lipids and lipoproteins, NMR analysis of low-molecular-weight metabolites provides the possibility to analysis the alteration in central metabolism following by the uptake of dietary phytochemicals. NMR-based profiles of biofluids can indeed detect amino acids, ketone bodies, metabolites related to glycolysis, the tricarboxylic acid cycle (TCA), lipolysis, the urea cycle, muscle metabolism, the Cori cycle, and oxidative stress.

However, so far dietary effects on central metabolism has been mainly evaluated in animal models due to their reduced metabolic inter-variability respect to that emerged from human metabolism studies [107].

Although NMR-based nutritional metabolomics is extensively applied to study human (or animal) metabolome as a function of nutritional status or as a function of a nutritional challenge [116,117,118,119], which could be the nutraceutical supplementation, its prospects are able to effectively monitor the human metabolic response to dietary interventions, without stressing nutraceuticals fate. In this context, further effort is needed to understand through NMR metabolomic approach how nutraceutical itself is metabolized and to assess the impact of other nutrient and non-nutrient compounds on its chemical structure and bioactivity preservation.

## 6. Conclusions

NMR-based metabolomics is a powerful platform for the study of nutraceuticals, offering possibilities at all of the levels of research in the field.

Its application in the search for new bioactive compounds or new sources of nutraceuticals is well established. While its use in the quality control of food, botanicals, and drugs is also a rather routine procedure, the applications to quality control of nutraceuticals still needs to be better established. However, this is a consequence of the scarce regulation on one side and of the limited clarity of the literature on the other side: it is often not exactly clear what really a nutraceutical is and what parameters are to be taken under consideration. The presence of a clear regulation on nutraceutical products is desirable, as well as a more in-depth knowledge of their fate in the human organism. Metabolomics research addressing this latest aspect is also increasingly growing. Of note, the suitability of NMR for high-throughput analysis of biological fluids is a clear advantage of this spectroscopic technique and an automated high-throughput serum and plasma NMR metabolomics platform is already available.

Undoubtedly, nutraceuticals research will benefit from the continuous technical improvement in NMR-based metabolomics. A big effort should be made in terms of standardization and of availability of public databases.

## Figures and Tables

**Figure 1 molecules-25-01444-f001:**
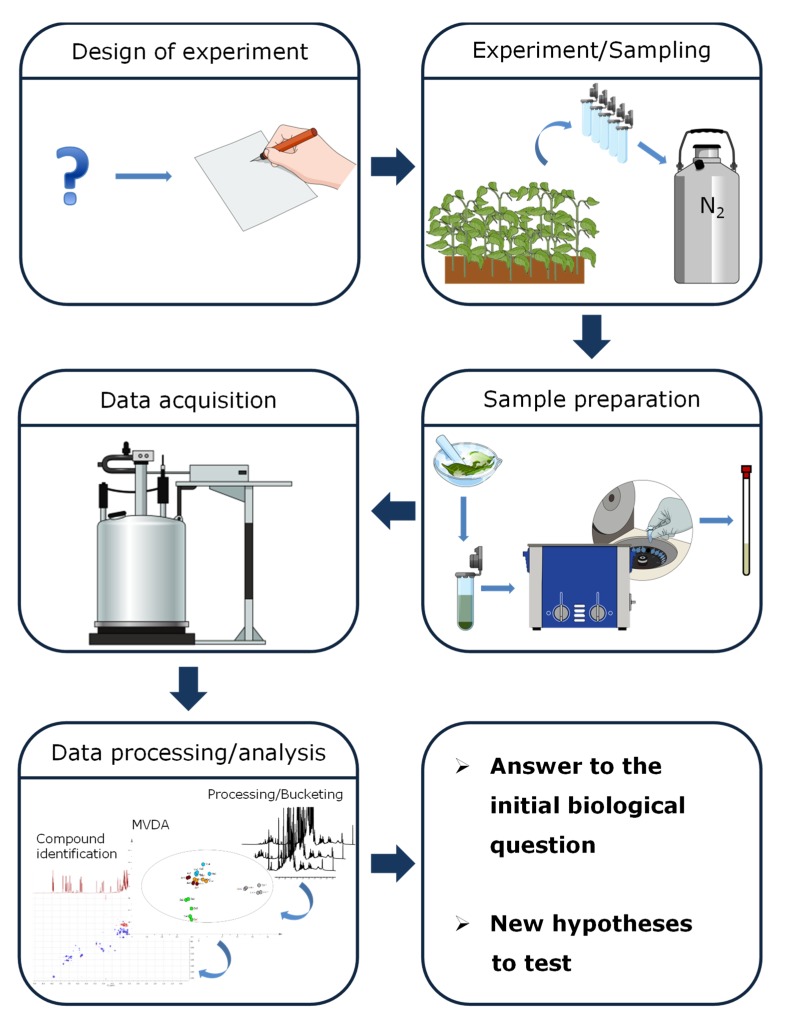
Pipeline of an NMR-based metabolomics experiment: in order to obtain results that are biologically and statistically significant a well-designed experimental approach is needed. This encompasses all the stages from sampling through data acquisition to data analysis (Figure created with mind the graph platform).

**Figure 2 molecules-25-01444-f002:**
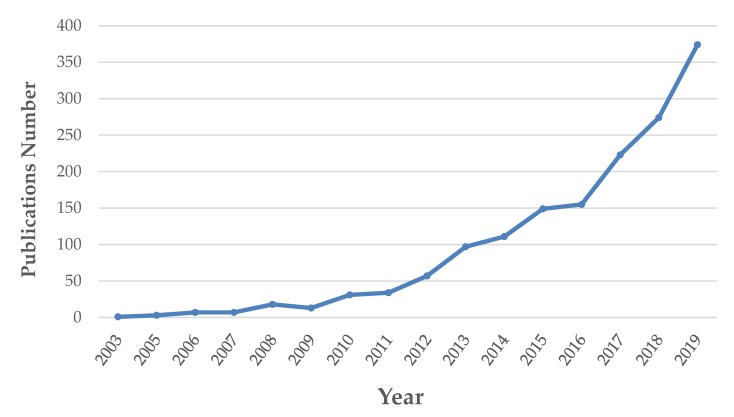
Increasing trend in nutraceutical publications obtained using the keywords “nutraceutical” and “metabolomics” from Scopus (https://www.scopus.com). At submission, the document count for 2020 is 84.

**Table 1 molecules-25-01444-t001:** Commonly used 2D-NMR (Nuclear Magnetic Resonance Spectroscopy) experiments.

Experiment	HO	HE	Description
COSY	√		Correlates proton signals belonging to geminal or vicinal hydrogen nuclei. Several variants are available. Among them, DQF-COSY, which (in principles) allows to detect also coupling constants.
TOCSY	√		Detects proton signals belonging to the same spin system.
HSQC		√	Detects carbon multiplicity and correlates carbon signals with directly bound protons.
H2BC		√	Correlates protons with the vicinal carbons, as long as these carbons are protonated.
HMBC		√	Allows the detection of long-range correlations. A variant of this is CIGAR-HMBC.
HSQC-TOCSY		√	Detects carbon multiplicity and the correlation of the attached proton signals belonging to the same spin system.

HO = homocorrelated experiment; HE = heterocorrelated experiment.

**Table 2 molecules-25-01444-t002:** Principal known nutraceuticals identified in extracts of different plant sources by 1D or 2D NMR-based metabolomics.

Compounds	Bioactivity [35,36,37,38,39,40,41,42,43,44,45]						Source Identified by NMR-Based Metabolomics
	Anticancer	Antioxidant	Anti- Inflammatory	Antibacterial	Antidiabetic	Cardiovascular Protection	
Carotenoids							
Lutein	√	√				√	*Diospyros kaki* L. [46]
Zeaxanthin	√	√				√	*Diospyros kaki* L. [46]
Phenolic compounds							
Curcumin	√	√	√	√	√		*Curcuma mangga* Valeton & Zijp [47]
Resveratrol	√	√	√	√			*Polygonum cuspidatum* Siebold & Zucc. [48]
Phenolic acids							
Caffeic acid	√	√			√		*Diospyros kaki* L. [46] *Musa* spp. [49] *Orthosiphon stamineus* Benth. [50] *Salvia hispanica* L. [51] *Withania somnifera* L. [52]
Chlorogenic acid	√	√			√		*Bougainvillea spectabilis* Willd. [53] *Coptis chinensis* Franch [54] *Coriandrum sativum* L. [55] *Cosmos caudatus* Kunth [56] *Salvia hispanica* L. [51] *Hypericum hircinum* L. [57] *Hypericum perforatum* L. [57] *Hypericum scruglii* Bacch., Brullo & Salmeri [57] *Lactuca sativa* L. [58] *Matricaria recutita* L [59] *Orthosiphon stamineus* Benth. [50] *Taraxacum officinale* F.H.Wigg. [60]
Ferulic acid	√	√			√		*Angelica* spp. [61] *Coptis chinensis* Franch [54] *Oryza sativa* L. [62] *Rubus coreanus* Miq. [63]
Gallic acid	√	√		√	√		*Bougainvillea spectabilis* Willd. [53] *Muntingia calabura* L. [64] *Oryza sativa* L. [62] *Orthosiphon stamineus* Benth. [50] *Phoenix dactylifera* L. [31] *Rubus coreanus* Miq. [63]
Flavonoids							
Flavanones							
Hesperidin	√	√	√	√		√	*Actinidia* spp. [65] *Bougainvillea spectabilis* Willd. [53]
Naringenin	√	√	√	√	√	√	*Lycopersicon esculentum* (Dunal) D’Arcy [24]
Flavones							
Apigenin	√	√	√	√			*Orthosiphon stamineus* Benth. [50]
Luteolin	√	√	√	√			*Orthosiphon stamineus* Benth. [50] *Papaver rhoeas* L. [60]
Flavonols							
Kaempferol	√	√	√	√	√	√	*Actinidia* spp. [65] *Bougainvillea spectabilis* Willd. [53] *Lycopersicon esculentum* (Dunal) D’Arcy [24]
Myricetin	√	√				√	*Vitis vinifera* L. [66]
Quercetin	√	√	√	√	√	√	*Cosmos caudatus* Kunth [56] *Hypericum hyrcinum* L. [57] *Hypericum perforatum* L. [57] *Salvia hispanica* L. [51]
Rutin	√	√	√			√	*Actinidia* spp. [65] *Bougainvillea spectabilis* Willd. [53] *Cosmos caudatus* Kunth [56] *Malus x domestica* Burch. [67] *Papaver rhoeas* L. [60]
Anthocyanidins							
Cyanidin	√	√			√	√	*Rubus coreanus* Miq. [63]
Catechins							
Epicatechin	√	√		√	√		*Hypericum perforatum* L. [57] *Hypericum scruglii* Bacch., Brullo & Salmeri [57] *Malus x domestica* Burch. [67] *Phoenix dactylifera* L. [31] *Rubus coreanus* Miq. [63]
Epicatechin gallate	√	√		√	√		*Diospyros kaki* L. [46]
Isoflavones							
Daidzein	√	√					*Muntingia calabura* L. [64] *Pueraria lobata* (Willd.) Ohwi [68] *Pueraria thomsonii* Benth. [68]
Genistein	√	√			√		*Salvia hispanica* L. [51]
Vitamins							
C (Ascorbic acid)	√	√	√				*Actinidia* spp. [58] *Capsicum annuum* L. [69,70] *Diospyros kaki* L. [46] *Lactuca sativa* L. [58] *Orthosiphon stamineus* Benth [50] *Rubus coreanus* Miq. [63]
E (α-tocopherol)	√	√	√				*Actinidia* spp. [65] *Diospyros kaki* L. [46] *Oryza sativa* L. [62]
E (γ-tocopherol)	√	√	√				*Muntingia calabura* L. [64]
Alkaloids							
Caffeine		√					*Coffea arabica* L. [71] *Coffea canephora* Pierre [71]
Berberine	√	√					*Coptis chinensis* Franch [54] *Mahonia aquifolium* (Pursh) Nutt. [72]
Trigonelline	√	√	√	√	√	√	*Actinidia* spp. [65] *Allium sativum* L. [73] *Bougainvillea spectabilis* Willd. [53] *Capsicum annuum* L. [69] *Cymbopogon schoenanthus subsp. proximus* (Hochst. ex A.Rich.) Maire & Weiller [74] *Cynara cardunculus* L. [75] *Coffea arabica* L. [71] *Coffea canephora* Pierre [71] *Cucumis melo* L. [76] *Cucurbita pepo* L. [77] *Diospyros kaki* L. [78] *Hypericum perforatum* L. [57] *Lycopersicon esculentum* (Dunal) D’Arcy [24] *Miscanthus × giganteus* J.M.Greef & Deuter ex Hodk. & Renvoize [79] *Pisum sativum* L. [80] *Papaver rhoeas* L. [60] *Taraxacum officinale* F.H.Wigg. [60] *Urtica dioica* L. *Withania somnifera* (L.) Dunal [52]
Diallylthiosulfinate							
Allicin	√	√	√	√		√	*Allium sativum* L. [73]

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
