# Peer review of "NMR-Based Plant Metabolomics in Nutraceutical Research: An Overview"

_molecules, 2020, doi:10.3390/molecules25061444_

Round 1

Reviewer 1 Report

Line 166   Curcurma longa,   change to   Curcuma longa,

Line 166-167  Use the complete scientific name Curcuma longa L., …

In Table 2 some scientific names of the plants are complete (i.e. with the name of the botanic) and others are incomplete, it’s better to introduce all the botanic names since without that can be confused and difficult to understand what plant the authors are talking about.  

Table 2 Curcuma manga [50]   change to Curcuma mangga [50]

Withania somnifera L. [55]   is in fact  Withania somnifera (L.) Dunal

Hypericum hyrcinum   change to   Hypericum hircinum L.

Ethical concerns are about the use of a study that refers to a non existing plant. Persicaria minus    This plant doesn’t exist, the reviewer did confirm the name in the original paper and the authors did copy the name correctly, but the problem remains, this plant doesn’t exist what exist is Persicaria minor (Huds.) Opiz that has a synonym Polygonum minus Huds. It’s better that the authors suppress this paper from the manuscript and not perpetuate the error. The paper is 31. Hussin, et al., NMR-Based Metabolomics Profiling for Radical Scavenging and Anti-Aging Properties of Selected Herbs. Molecules, 2019. 24(17).

Rubus coreanus Miquel   change to   Rubus coreanus Miq.

References

Some problems were detected in References specially in the titles, so correct in accordance.

Example:

  1. Frédérich, M., et al., Quality Assessment ofPolygonum cuspidatumandPolygonum multiflorumby1H NMR Metabolite Fingerprinting and Profiling Analysis. Planta Medica, 2010. 77(01), 81-86.

Author Response

Line 166   Curcurma longa,   change to   Curcuma longa,

Line 166-167  Use the complete scientific name Curcuma longa L., …

In Table 2 some scientific names of the plants are complete (i.e. with the name of the botanic) and others are incomplete, it’s better to introduce all the botanic names since without that can be confused and difficult to understand what plant the authors are talking about.  

Table 2 Curcuma manga [50]   change to Curcuma mangga [50]

Withania somnifera L. [55]   is in fact  Withania somnifera (L.) Dunal

Hypericum hyrcinum   change to   Hypericum hircinum L.

Ethical concerns are about the use of a study that refers to a non existing plant. Persicaria minus    This plant doesn’t exist, the reviewer did confirm the name in the original paper and the authors did copy the name correctly, but the problem remains, this plant doesn’t exist what exist is Persicaria minor (Huds.) Opiz that has a synonym Polygonum minus Huds. It’s better that the authors suppress this paper from the manuscript and not perpetuate the error. The paper is 31. Hussin, et al., NMR-Based Metabolomics Profiling for Radical Scavenging and Anti-Aging Properties of Selected Herbs. Molecules, 2019. 24(17).

Rubus coreanus Miquel   change to   Rubus coreanus Miq.

Response: Thank you for your comments and especially for highlighting the issue with the reference by Hussin et. al. We corrected this issue and all the botanicals name in the table 2 and in the text.

References

Some problems were detected in References specially in the titles, so correct in accordance.

Example:

  1. Frédérich, M., et al., Quality Assessment ofPolygonum cuspidatumandPolygonum multiflorumby1H NMR Metabolite Fingerprinting and Profiling Analysis. Planta Medica, 2010. 77(01), 81-86.

Response: References were corrected.

Reviewer 2 Report

The manuscript entitled NMR-based metabolomics to characterize plant derived nutraceuticals provides an overview of the current use of NMR metabolomics in this very topical research field, and cover a very broad range of application areas, from discovery of nutraceuticals in new plant sources, quality control and optimisation, to regulatory issues of nutraceuticals. As the remit of the review appears very broad, the review article does not necessary provide a comprehensive review of the current literature in some of the sections. I have a few suggestions:

1.       Table 2 is a very handy summary of the example literature of NMR applications. These example studies could be discussed in the text in greater details.

2.       Is NMR a suitable technique for covering the full range of nutraceuticals compound classes of interest. The authors should try to address this in the review article.

3.       Does the authors agree with Steven Zeisel that nutraceuticals could be defined ‘as those diet supplements that deliver a concentrated form of a presumed bioactive agent from a food, presented in a nonfood matrix, and used to enhance health in dosages that exceed those that could be obtained from normal foods.’ (DOI: 10.1126/science.285.5435.1853)

4.       Line 140 – 142. The authors mentioned that the global nutraceutical market will reach $336 billion by 2023. The source for this research should be cited.

5.       Could the authors comment on whether working at higher magnetic field strength (e.g. 1.2GHz) – a trend in metabolomics research in the last 2 decades – is likely to help overcome sensitivity issues and improve the coverage of low abundant phytochemicals in future studies?

6.       What does NMR stand for? Although this should be obvious for many readers, this should still be mentioned in the article.

7.       References 16 and 33 (Markley et.al) are duplicated in the Reference list. Please correct

Author Response

Reviewer 2

The manuscript entitled NMR-based metabolomics to characterize plant derived nutraceuticals provides an overview of the current use of NMR metabolomics in this very topical research field, and cover a very broad range of application areas, from discovery of nutraceuticals in new plant sources, quality control and optimisation, to regulatory issues of nutraceuticals. As the remit of the review appears very broad, the review article does not necessary provide a comprehensive review of the current literature in some of the sections. I have a few suggestions:

  1. Table 2 is a very handy summary of the example literature of NMR applications. These example studies could be discussed in the text in greater details.

Response: thank you for your comment, However, we believe that giving a detailed discussion of these works would only make the review too long, without adding any further important information.

  1. Is NMR a suitable technique for covering the full range of nutraceuticals compound classes of interest. The authors should try to address this in the review article.

Response: Please see lines 112-116. The concept was already there, but it has been further developed now

  1. Does the authors agree with Steven Zeisel that nutraceuticals could be defined ‘as those diet supplements that deliver a concentrated form of a presumed bioactive agent from a food, presented in a nonfood matrix, and used to enhance health in dosages that exceed those that could be obtained from normal foods.’ (DOI: 10.1126/science.285.5435.1853)

Response: We do and we added the reference to the text.

  1. Line 140 – 142. The authors mentioned that the global nutraceutical market will reach $336 billion by 2023. The source for this research should be cited.

Response: The source was cited

  1. Could the authors comment on whether working at higher magnetic field strength (e.g. 1.2GHz) – a trend in metabolomics research in the last 2 decades – is likely to help overcome sensitivity issues and improve the coverage of low abundant phytochemicals in future studies?

Response: Thank you for raising this point. Some lines (130-135) on this were added to the text.

  1. What does NMR stand for? Although this should be obvious for many readers, this should still be mentioned in the article.

Response: Added

  1. References 16 and 33 (Markley et.al) are duplicated in the Reference list. Please correct

Response: The replicated reference was removed

Round 2

Reviewer 1 Report

Please add  access date of websites.